A review of advancements in the theory and characterization of soil macropore structure

Cao Yanwen
Zhang Kai zhangkai@cumtb.edu.cn
Liu Shuyu
Wang Yilin
School of Chemical and Environmental Engineering, China University of Mining and Technology- Beijing , Beijing , China
Adhikari Kabindra
Electronic publication date: 2024 Nov 14
Publication date: 2024
Volume: 12
Electronic Location ID: e18442
Received 2024 Apr 30; Accepted 2024 Oct 11
Copyright: ©2024 Cao et al.
Copyright year: 2024
Copyright holder: Cao et al.
License: This is an open access article distributed under the terms of the Creative Commons Attribution License, which permits unrestricted use, distribution, reproduction and adaptation in any medium and for any purpose provided that it is properly attributed. For attribution, the original author(s), title, publication source (PeerJ) and either DOI or URL of the article must be cited.
License URL: https://creativecommons.org/licenses/by/4.0/

Keywords: Soil macropore, Pore structure characterization, Quantification, Computed tomography, Mercury intrusion porosimetry, Bibliometric analysis

Funding: The National Nature Science Foundation of China 42177037 The Fundamental Research Funds for the Central Universities 2023JCCXHH02 National Key Research and Development Program of China 2022YFF1303304 This work was supported by the National Nature Science Foundation of China (42177037), the Fundamental Research Funds for the Central Universities (2023JCCXHH02), and the National Key Research and Development Program of China (2022YFF1303304). The funders had no role in study design, data collection and analysis, decision to publish, or preparation of the manuscript.

==============================
Soil has an intricate pore structure, with the macropore structure serving as a pivotal component of gas exchange, liquid flow, and solute transport within the soil matrix. Scientifically understanding the macropore structure is important for the exploration of soil ecological functions. Drawing upon research literature from diverse nations, this study presents a comprehensive summary of past research findings, thoroughly examining the definition and taxonomy of soil macropores. Additionally, we delve into the factors that affect macropore structure, the effects of macropores on soil ecological functions, and the indicators that describe the nature of macropores. Furthermore, the principles, suitability conditions, benefits, and limitations of the widely utilized techniques for characterizing soil macropore structure were outlined, including staining observation, soil water retention curve, mercury intrusion porosimetry, gas adsorption, nuclear magnetic resonance, scanning electron microscopy, and computed tomography (CT). A comparative analysis was conducted to assist scholars in comprehending the theory of soil macropores, facilitating the selection of suitable macropore structure characterization techniques, and enabling the development of collaborative application modes among these techniques. Using the Web of Science core collection database, bibliometric software such as VOSviewer and CiteSpace were employed to explore the research hotspots and trends in CT, which is currently the most popular technique for characterizing soil macropore structure. This analysis offers insights into the future prospects of CT in the field of soil macropore structure characterization techniques. This article is of particular interest to readers engaged in the characterization of soil macropores.

Introduction

Soil is a complex porous medium, and the spaces between soil particles are referred to as soil pores. Soil pores are frequently classified into distinct categories. In general, macropores are defined as those with large diameters. Micropores are defined as those with small diameters, and mesopores are pores with diameters between macropores and micropores. However, the specific diameter limits are subject to variation depending on factors such as the research objectives. A unified standard for the definition of soil pores has yet to be established by the academic community. Soil macropores, a significant component of soil pore structure, serve as primary spaces for water and air within the soil. They also act as channels for the transportation of liquid- and vapor-phase substances and play crucial roles in soil hydrological processes, microbial activity, and plant growth (Pierret et al., 2002). The macropore structure of soil is not static and can be influenced by both anthropogenic and natural factors (Zhang et al., 2023). For instance, agricultural practices, such as plowing, often loosen the surface soil structure and increase porosity, whereas mechanical compaction in open-pit mines has the opposite effect (Feng et al., 2019; Thotakuri et al., 2024). The activity of organisms, such as earthworms, present in natural soils results in the transition of the soil pore structure from a relatively homogenous orbicular structure to an elongated one (Ma et al., 2021). These changes in the soil macropore structure can further affect gas exchange, liquid flow, and solute transport in the soil, thereby modifying its ecological function (Jarvis, 2020). The ecological function of soil pore space, refers to the effects of soil pore space on ecosystems, including the effects of soil pore space on soil water circulation, gas exchange, nutrient cycling, biodiversity and environmental protection. Given the significance of the soil macropore structure in determining the ecological function of soil, it is imperative to investigate its characteristics. At present, there is no unified understanding of macropore theory within the academic community, which has a significant impact on the work of scholars engaged in macropore research.

Quantitative characterization is essential for analyzing the soil macropore structure, which serves as the foundation and prerequisite for understanding the ecological functions of the soil pore structure. In contrast to soil particles, soil pores lack discernible physical boundaries, rendering their identification, characterization, and assessment a challenging endeavor (Nimmo, Katuwal & Lucas, 2023). Scholars have utilized various techniques, including staining observation (SO), soil water retention curve (SWRC), mercury intrusion porosimetry (MIP), gas absorption (GA), nuclear magnetic resonance (NMR), scanning electron microscopy (SEM), and computed tomography (CT), to quantitatively characterize soil pore structure. On this basis, they delved into the ecological functions of soil pore structure. For instance, Wang et al. (2020a) and Wang et al. (2020b) used MIP and selected three microporosity parameters–total pore volume, median pore diameter, and average pore diameter–to quantitatively describe the changes in soil microporosity before and after freeze-thaw consolidation. Kong et al. (2022) used NMR to determine the pore size distribution of freeze-thawed soft soils. They examined how factors such as freezing time, freezing and thawing pressures, and freezing temperatures affect the micropore structure of soils. Cui & Jia (2013) utilized SEM to investigate changes in the microstructural characteristics of soil before and after centrifugal model tests of high-rise building complexes. They considered the pore ratio, equivalent diameter, circularity, shape factor, orientation angle distribution, and fractal dimension of pore morphology. Sun et al. (2021) employed CT in combination with fractal theory to assess soil pore diameter, circularity, three-dimensional fractal, connectivity density, and porosity in biochar-amended saline soils. They also examined the relationship between the three-dimensional fractal characteristics of pores and soil hydraulic properties. In conclusion, a multitude of techniques exist for the characterization of soil macropores, each with distinct principles, advantages, and disadvantages. These factors not only influence the definition of macropore indicators but also the efficacy of macropore characterization. Scholars have employed different techniques to quantitatively assess the soil pore structure to obtain distinct parameters, leading to distinct conclusions. It is therefore important to be aware of the techniques that are most applicable to the different aspects of macropore. A comprehensive review of the advantages, disadvantages, and applicability of macropore characterization techniques is lacking, impeding scholars’ ability to select appropriate soil pore structure characterization techniques for their research topics.

In light of the aforementioned considerations, the purpose of this article is to help the readers understand the theory of macropores and research advances, and to select appropriate techniques for characterizing soil macropores, with the aim of addressing the current research gaps in this domain. Moreover, a bibliometric analysis of research hotspots and trends in CT, the most promising pore characterization technique, were be presented. This article should be of particular interest to readers in the field of soil macropores, especially soil macropores characterization.

Survey Methodology

The objective of this review is to provide an overview of the recent research progress in the theory and characterization of soil macropores. Accordingly, all efforts are merged, discussed, and compared to provide broad and accurate knowledge of all developed techniques and the corresponding findings. First, the theories of macropores, including the definition and classification of soil macropores, factors that affect macropore structure, effects of macropores on soil ecological functions, and indicators that describe the nature of macropores, are introduced and discussed. Subsequently, we summarize the currently prevalent intrusive techniques such as SO, SWRC, MIP, and GA, and non-intrusive techniques such as SEM and CT. The principles, conditions of application, advantages, and disadvantages of each technique are compared and analyzed. Furthermore, the feasibility of combining CT with other techniques is investigated.

Finally, using the Web of Science (WOS) Core Collection database (1998–2023), we employed bibliometric software such as VOSviewer and CiteSpace to conduct a comprehensive analysis of the research trends of CT, a popular technique in soil macropore characterization. This analysis will aid in the selection and in-depth research of soil macropore characterization techniques in the future.

Theory of soil macropores

Definition and classification of macropores

The concept of soil macropores is subjective. The most straightforward and commonly used method for defining macropores is based on the equivalent diameter (ED). However, there is still no consensus in the academic community regarding the classification of macropore diameters. Consequently, the pore definitions provided in different studies vary significantly, as summarized in Table 1. Yang et al. (2018) used CT to investigate the impact of different soil amendment methods on soil pore properties and pore distribution. They categorized soil pores into two categories: macropores (>1 mm in diameter) and smaller pores (0.13–1.0 mm). Kong et al. (2020) employed NMR to ascertain the pore size of soil subjected to freeze-thaw conditions and classified the pores into tiny pores (0−0.2 µm), medium pores (0.2–10 µm), and large pores (>10 µm). Gao et al. (2018) observed loess pores with SEM and subsequently classified the pores into four categories: micropores (ED <2 µm), minipores (2 µm <ED <8 µm), mesopores (8 µm <ED <32 µm), and macropores (D >32 µm). In order to examine the influence of salt content and salt type on pore structure, Wang et al. (2020a) and Wang et al. (2020b) used MIP to determine the pore size distribution, which was classified into micropores (<0.04 µm), small pores (0.04−0.4 µm), and mesopores (0.4-4 µm). Overall, pore classification thresholds and terminology vary among researchers owing to differences in the research objectives, methods, and equipment used. Nevertheless, it is common to classify soil pores into three categories: large, medium, and small.

Table 1 Comparison of pore definitions by various scholars based on equivalent diameter (ED).

Classification	Range of ED	References	
Small pores	30 µm ≤ ED ≤ 100 µm	Zhou et al. (2010)	
Medium pores	100 µm ≤ ED ≤ 500 µm	
Large pores	ED >500 µm	
Micropores	ED <2 nm	Zhang & Kong (2013)	
Mesopores	2 nm ≤ ED ≤ 50 nm	
Macropores	ED >50 nm	
Micropores	ED <3µm	Hashemi et al. (2015)	
Mesopores	3 µm ≤ ED ≤ 30µm	
Macropores	ED >30µm	
Micropores	0 µm ≤ ED ≤ 1µm	Li & Li (2017)	
Small pores	1 µm ≤ ED ≤ 4 µm	
Mesopores	4 µm ≤ ED ≤ 16 µm	
Macropores	ED >16 µm	
Smaller pores	130 µm ≤ ED ≤ 1000 µm	Yang et al. (2018)	
Macropores	ED >1000 µm	
Micropores	ED <2 µm	Gao et al. (2018)	
Minipores	2 µm <ED <8 µm	
Mesopores	8 µm <ED <32 µm	
Macropores	ED >32 µm	
Small pores	ED <8µm	Zhang et al. (2020)	
Medium pores	8 µm <ED <13 µm	
Large pores	ED >13 µm	
Mesopores	0.4 µm <ED <4 µm	Wang et al. (2020a) and Wang et al. (2020b)	
Macropores	ED >4 µm	
Tiny pores	0 µm <ED <0.2 µm	Kong et al. (2020)	
Medium pores	0.2 µm <ED <10 µm	
Large pores	ED >10 µm	
Ultramicro pore	d < 0.1µm	Wang et al. (2022)	
Micro pore	0.1 µm <ED <2 µm	
Small pore	2 µm <ED <10 µm	
Medium pore	10 µm <ED <20 µm	
Macro pore	ED >20 µm	
Micropores	ED <30 µm	Peng et al. (2023)	
Mesopores	30 µm ≤ ED ≤ 75 µm	
Macropores	75 µm ≤ ED ≤ 100 µm	
Ultramacropores	ED >100 µm	

Although the definition of macropores in terms of pore size remains elusive, their classification based on origin and morphology has gained widespread acceptance within the academic community (Liu, Qu & Ying, 2001). Soil macropores can be classified based on their origin, with biological pores referring to plant root pores and soil animal burrows, abiotic pores resulting from freeze-thaw and wet-dry cycles, and human activities such as transportation and cultivation, and natural macropores forming based on soil textural factors. Macropores can also be categorized by their morphology into holes, fissures, cracks, tubes, and irregular macropores (Zhang, Xu & Pei, 2012). Holes have a spherical structure, cracks have a ratio of long to short axes greater than 10, fissures are smaller than cracks, and tubes have an approximate cylindrical shape. Studies have shown that the shape of macropores is correlated with their origin. For example, macropores formed by plants and animals are usually tubular, whereas cracks or fissures are often formed by physical processes, such as collapse, freeze-thaw, and wet-dry cycles. The macropores formed by the soil aggregates are primarily irregular. Zong et al. (2015) discovered that macropores in soil formed by decaying roots often assume tubular or elongated shapes. However, pores created by repeated wet-dry cycles are smaller, more random, and less continuously distributed. Other researchers have proposed alternative nomenclatures for the classification of macropore morphologies. For example, Zong et al. (2015) classified pores into regular (0.5–1.0), irregular (0.2–0.5), and elongated (0.0–0.2) based on the range of the shape factor. Dhaliwal & Kumar (2022) categorized pores into equant, oblate, triaxial, prolate, planar, acicular-planar, and acicular based on the ratio of the intermediate to large axis and the short to intermediate axis. Although the shape factor is a useful parameter for pore classification, there is no fixed set of standard thresholds for classification; therefore, thresholds need to be tailored to the experience and actual pore distribution patterns of each research object (Li et al., 2019a; Li et al., 2019b). The classification and definition of macropores are crucial for better understanding the characteristics of different types of macropores and are essential prerequisites for the study of soil macropores.

Factors affecting macropore structure

The structure of soil macropores is affected by various factors. Anthropogenic factors, such as coal mining, mechanical compaction, and tillage management, play a significant role, though non-anthropogenic factors, including animals, plants, soil types, and wet-dry cycles, also exert an influence, as shown in Table 2. These factors can lead to significant differences in the macropore structure of soil. In addition to the description in the previous section that the shape of macropores can be affected by their origin, Dhaliwal & Kumar (2022) found that the emergence of distinct macropore shapes is linked to the soil type and management practices. Zhao, Hu & Li (2020) determined that several factors, including plant roots, freeze-thaw cycles, temperature fluctuations, and precipitation patterns, can affect pore morphology. Moreover, they observed that the higher the organic matter content of the soil, the more irregular the pore shape. The functioning of macropores can also be affected by many factors, Coal mining and mechanical compaction have destructive effects on the macropores, diminishing their multifunctionality. Conversely, plant roots typically enhance soil pore structure and increase porosity and pore connectivity. The behavior of different soil animals also vary in their effects on macropores. Furthermore, freeze-thaw and wet-dry cycles contribute to the increase in macropore number and complexity.The researchers can choose the most relevant macropore properties for their study purposes. For example, if the objective is to investigate the impact of biological factors on soil macropores, the focus should be on macropore connectivity. Instead, if anthropogenic factors, such as mechanical compaction, are the subject of study, attention should be given to the macropore shape and diameter.

Table 2 Evolution of research on the factors affecting soil macropore structure.

Factors	Findings	References	
Anthropogenic factors	coal mining	① Open-pit coal mining leads to a decrease in soil macroporosity, resulting in a more uniform distribution of soil macropores and a reduction in the coefficient of variation of their volume.	Wang, Qin & Bai (2018)	
② The increase in the extent and spatial variability of macropores diameter distribution is a result of subsidence caused by underground coal mining.	Li et al. (2022)	
Mechanical compaction	① The mining and drainage activities have a profound impact on the soil pore structure, leading to an increase in the number of ineffective pores within the soil.	Wang et al. (2016)	
② Mechanical compaction results in a significant reduction in soil macroporosity, a decrease in the number, equivalent diameter, and versatility of macropores, and an increase in macropore roundness.	Min, Jiao & Li (2020)	
Tillage management	① The macroporosity of organic agriculture is superior to that of conventional management, both in terms of quantity and quality.	Wang et al. (2021)	
② The total soil porosity and porosity of larger pore diameters are greater in ridge tillage compared to no-tillage.	Liu et al. (2023)	
non-anthropogenic factors	Animals	① Animal trampling can lead to soil sloughing, which disrupts the soil’s structure and diminishes its macroporosity.	Dhaliwal & Kumar (2022)	
② The ingestion and excretion of worms can result in a decrease in pore size and connectivity.	Holden (2009)	
③ The ingestion and burrowing behavior of earthworms can enhance macroporosity, increase the number of macropores, and alter the morphology of soil macropores.	Ma et al. (2021)	
Plants	① The macroporosity, length density, and node density of the pasture soil exceeded those found under row crops.	Luo, Lin & Li (2010)	
② The macropores in the soils of mixed forests exhibit larger diameters, surface areas, network densities, length densities, and smaller average curvatures compared to other soils.	Meng et al. (2017)	
③ The forest soil macropores networks exhibit remarkable connectivity, vertical continuity, linearity, and poor vertical orientation.	Zhang et al. (2017)	
Soil types	① In brown soil regions, the vertical continuity of macropores is high, while in karst artificially restored forest areas, the horizontal development and fragmentation of macropores are more prevalent.	Kan et al. (2021)	
② Macroporosity is influenced by the suitability for plant growth and biota activity of different soils.	Luo, Lin & Li (2010)	
Freeze-thaw and wet-dry cycles	① The formation of smaller macropores in the topsoil is promoted by wet-dry cycles, resulting in a decrease in the average macropores diameter.	Budhathoki et al. (2022)	
② After freeze-thaw cycles, the macropores with smaller pore diameters tend to increase, while those with larger pore diameters tend to decrease. Moreover, the more freeze-thaw cycles occur, the greater the number of macropores and the larger the effective porosity.	Jiang et al. (2021)	
③ As the number of freeze-thaw cycles increases, the soil macropores structure becomes increasingly porous and intricate.	Liu et al. (2021)	

Indicators of macropores and effects of macropores on soil ecological functions

The properties of macropores can be described using various indicators (Schlüter & Vogel, 2022). These indicators can be divided into two categories: those that describe macropore morphology and those that characterize macropore functions. The indicators of macropore morphology include parameters such as pore equivalent diameter, Ferret diameter, fractal dimension, roundness, sphericity, shape factor, and tortuosity. These indicators provide information on the shape and structure of the macropores. Indicators that characterize the macropore function include parameters such as pore number, porosity, pore size distribution, maximum pore area, pore-throat ratio, Euler number, inclination angle, surface area density, network density, and junction density. Table 3 presents the definitions and calculations of some commonly used macropore indicators.

Table 3 Definition or calculation of selected soil macropore indicators.

Categories	Indicators	Definition or calculation method	
Indicators describe macropore morphology	Equivalent diameter	In 2D, equivalent diameter is the diameter of the disk of the same area. In 3D, equivalent diameter is the diameter of the sphere of the same volume.	
Ferret diameter	Feret diameter is a one-dimensional measurement that estimates how “wide” an object is in a given direction.	
Fractal dimension	Fractal dimension indicates the self-similarity of the pores. It can be converted using the relationship between pore perimeter and area.	
Roundness	Roundness is the degree of closeness of the pore morphology to a standard circle. Its value is between 0 and 1. The closer it is to 1, the closer the pore is to a circle.	
Sphericity	Sphericity is the degree of closeness of the pore morphology to a standard sphere. Its value is between 0 and 1. The closer it is to 1, the closer the pore is to a sphere.	
Shape factor	Shape factor is a dimensionless parameter used to quantify the pore geometry. The calculation and formulae employed to determine this parameter vary across different studies.	
Tortuosity	Tortuosity indicates the bending degree of macropores. It can be calculated as the ratio of actual length of macropores to the length of straight lines.	
Indicators characterize macropore functions	Porosity	Porosity is defined as the ratio of the volume of the pore space to the total volume of the soil.	
Pore size distribution	Pore-size distribution refers to the relative abundance of each pore size within a defined volume of soil.	
Pore-throat ratio	Pore-throat ratio is defined as the ratio of the diameter of the pore to that of the throat.	
Euler number	The Euler number is calculated according to the following formula: E = N − L + O . The number of completely closed and isolated pores is represented by N, the number of annular pores (redundantly connected or closed) is represented by L, and the number of pores is represented by O.	
Inclination angle	Inclination angle can be calculated as the angle between vertical direction of macropores and ground plane.	
Surface area density	Surface area density refers to the surface area of macropores per unit volume soil.	
Network density	Network density refers to the number of macropore per unit volume soil.	
Junction density	Junction density refers to the number of macropore nodes per unit volume soil.	

Soil macropores play a pivotal role in ecosystems and exert profound effects on the ecological functions of soil (Keller et al., 2013). These effects are primarily observed in the hydrological cycle, soil quality, and biodiversity (Li et al., 2023). Macropores, the main space within the soil that retains water and gases, facilitate the diffusion and transportation of water, gases, and solutes. Despite comprising only 0.1–5% of the soil volume, macropores serve as the preferred channels for soil water infiltration and evaporation. They can conduct approximately 90% of water flux, significantly influencing the rate, amount, and path of soil water transport (Hayashi, Ken’Ichirou & Mizuyama, 2006). Furthermore, soil solutes migrate within the preferential channels formed by macropores along with the movement of water, influencing the content and distribution of nutrients in the soil (Zhang et al., 2022b). More importantly, soil macropores provide essential living spaces and oxygen for plants, animals, and microorganisms within the soil. This not only impacts the reproduction of biological populations and the differentiation of ecological niches but also plays a significant role in maintaining biodiversity.

The macropore indicators is closely associated with the functional behavior of macropore (Alessandrino et al., 2023). For instance, an elevated porosity level is associated with enhanced aeration and drainage, although it may also result in a diminished water-holding capacity. The functional roles of pores of different sizes in soil are distinct. Macropores of a larger size are primarily responsible for facilitating the transportation of air and water, whereas those of a medium and smaller size are more involved in the processes of water retention and nutrient cycling (Ajayi et al., 2021). The functional behavior of macropores is also influenced by their connectivity, which can be quantified using Euler number and pore-throat ratio. Well-connected macropores can form effective pathways for gas and water, enhancing the overall performance of the soil. In contrast, poorly connected macropores may become isolated “dead pores”, contributing minimally to soil function. Scholars often select specific indicators to describe the properties of macropores quantitatively based on their research objectives. Feng et al. (2020) investigated the effect of mechanical compaction on the structural properties of soil macropores by characterizing the macroporosity, macropore number, area, volume, and connectivity. In another study, Kan et al. (2021) characterized the diameter, volume, surface area, and distribution of macropores in different types of soils and then calculated the macropore angle, tortuosity, actual length, and curvature to understand their relationship with water content. Li et al. (2019a) and Li et al. (2019b) analyzed loess permeability by considering factors such as porosity, pore diameter, tortuosity, and fractal dimensions. By selecting appropriate indicators, researchers can effectively describe the morphology and properties of macropores and understand their behavior in different environments.

Characterization Techniques for Soil Macropore Structure

Intrusive techniques

Intrusive techniques use gases or liquids to penetrate soil pore systems. These techniques indirectly calculate macropore parameters using various methods such as SO, SWRC, MIP, GA, and NMR. However, these techniques significantly disrupt soil and pore structure.

Staining observation

SO involves directly dyeing soil in the field or laboratory, taking photographs, processing, and analyzing digital images using image-processing techniques. This method is intuitive and is suitable for large-scale studies. The solution flows through the pore space in the soil, and the stained area represents the existence of the pore space. The more connected the pore space in depth, the deeper the staining. Liu et al. (2012) stained and observed the pore structure of soil profiles using brilliant blue. They characterized the volume of the macropores by calculating the stained area ratio. This allowed them to compare and analyze the soil macropore structure and its origin under different vegetation cover types. Additionally, SO allows for the observation of soil macropore flow paths and morphological features, making it a useful tool for characterizing and discriminating preferential flows. For example, Wang & Zhang (2011) used a combined tracer of dye and iodine to characterize macropore structure and heterogeneous soil water flow patterns. Dye staining was used to visualize the macropore structure, whereas iodine staining was used to visualize the non-uniform flow patterns.

Although convenient and efficient, use of SO for determining soil macropores is limited to soils with good permeability. This is because soils with good permeability facilitate the diffusion of the dye, producing easily observable staining patterns that provide information on the number of macropores and the type of pore flow. However, dyes can only diffuse through the soil matrix to a certain extent, making it challenging to accurately measure the number and area of macropores in soils with high pore density and poor permeability. Moreover, SO can cause significant damage and disturbance to soil macropores, preventing replication of the experiment at the same site. Furthermore, soil particles may adsorb dye, leading to an incomplete observation of preferential flow paths. Additionally, certain dyes may be unsafe and could potentially contaminate the soil (Filipovic et al., 2020). More importantly, macropore information obtained through SO has limited accuracy. Although it provides an approximate spatial distribution of macropores, it cannot quantitatively characterize morphological parameters, such as pore diameter, volume, and roundness, or functional parameters, such as pore connectivity and pore number.

Soil water retention curve, mercury intrusion porosimetry, and gas absorption

In addition to the macropore parameters derived from images, further characteristics, such as the distribution and quantity of macropores, can be determined by analyzing the curves. Soil water retention curve (SWRC), mercury intrusion porosimetry (MIP), and gas absorption (GA) are techniques commonly used for this purpose.

The use of measured SWRCs to obtain the soil pore distribution status is one of the most commonly employed and well-established methods in the study of soil pore distribution. A SWRC is a curve that indicates the capacity of a soil to hold and supply water. These curves can be divided into two categories: moisture absorption curves and dehumidification curves. By determining the SWRC, a series of values pertaining to soil water content and soil suction can be obtained. In the event that the pore structure is deemed to be regular, the soil pore size and pore size distribution can be indirectly inferred from the empirical formula between pore suction and effective pore radius. The water suction in large pores is relatively low, whereas that in small pores is high. Consequently, if there are a greater number of large pores in the soil, the SWRC will be relatively flat with a small slope. Conversely, if there are a greater number of small pores in the soil, the curve will be relatively steep with a large slope. Iiyama (2016) investigated the impact of drying conditions on the pore structure of soil through an in situ SWRC test. De Oliveira, Cássaro & Pires (2021) employed the traditional water use capacity function (first order derivative of SWRC) to estimate soil PSD.

The measurement of SWRC is frequently conducted in a laboratory setting rather than directly in situ. This is due to the inherent limitations of in situ measurements of SWRC, which are constrained by the natural wet/dry cycle. This cycle prevents soils from reaching a state of complete saturation or extreme dryness, thereby narrowing the scope of in situ SWRC measurements. Furthermore, there is frequently a discrepancy between laboratory and in situ measurements, and the application of laboratory-derived SWRC data to in situ conditions may result in errors. The process of laboratory measurement of SWRC typically necessitates the pre-treatment of soil samples, including water saturation and drying. These operations may induce alterations to the natural pore structure of the soil, thereby impeding the non-destructive characterization of the pores. The SWRC primarily reflects the water retention capacity of soil pores, simplifying the spatial distribution of the pore structure and the morphological characteristics. However, it is not capable of quantifying all the characteristics of soil. This is because it does not account for the characteristics of the pore space, such as the shape of the pore space, connectivity, and so on.

MIP is a precise and effective quantitative analysis technique of the size and distribution of pores in rock and soil masses. This relies primarily on the applied pressure to overcome the surface tension of the soil samples, allowing mercury to enter the macropore structure of the soil. Subsequently, the size and spatial distribution of macropores can be inferred indirectly based on the applied external pressure; therefore, MIP can be used to investigate the relationship between the macropore characteristics and macroscopic mechanical properties of soil samples. Compared to SO, MIP is simpler to operate. During mercury compression tests, only the inlet pressure and volume were measured and recorded. The soil pore parameters can then be calculated and analyzed using the appropriate formulas. MIP is frequently used to assess the porosity and pore size distribution of materials that feature open and interconnected pores (Wang et al., 2020a; Wang et al., 2020b). By meticulously analyzing the mercury intrusion and extrusion curves (Fig. 1C), comprehensive insights can be gained into the distribution patterns, connectivity, and permeability of macropores across the entire soil sample. Key pore characteristics such as total pore volume, average porosity, fractal dimension, and specific surface area can be quantitatively determined. MIP offers advantages in terms of wide measurement range and good repeatability of measurement results (Giesche, 2006). Unlike other pore characterization techniques, which have limited scale ranges, MIP can effectively detect pores across the entire range, particularly small pores. Consequently, MIP is indispensable for nanoscale pore studies.

Figure 1 Mercury intrusion porosimetry instrument and related images.

(A) Mercury piezometer (Yan, Fang & Zhang, 2011); (B) vacuum freeze-dryer (Yan, Fang & Zhang, 2011); (C) mercury intrusion and extrusion curves (Wang et al., 2020b). Source credit (A–B) Yingguang Fang.

For technical reasons, the MIP should be performed on completely dried soil samples. Therefore, sample pretreatment is essential for removing pore water and ensuring a smooth, bubble-free surface, minimizing the impact of mercury pressure on the soil structure (Stoltz, Cuisinier & Masrouri, 2012; Ma et al., 2020). However, the pretreatment process can alter the original macropore structure of the soil sample (Yang, Li & Nian, 2023). The primary limitation of MIP is its inability to identify closed macropore structure and accurately determine the geometric characteristics of isolated macropores. During the experimental process, the forced intrusion of non-wetting liquid mercury at high pressure can disrupt the macropore structure of the soil, leading to the deformation of the macropores and influencing the cumulative intrusion volume of mercury (Zhang et al., 2020). This can distort the detected macropore structure and result in inaccurate measurements of parameters, such as macroporosity. Additionally, pores located behind the neck of the bottle are prone to the ink-bottle effect, where mercury remains trapped in these pores upon withdrawal. Consequently, these pores may be misclassified as smaller, underestimating the total porosity (Zhang & Kong, 2013; Hashemi et al., 2015; Wang et al., 2020a; Wang et al., 2020b). Moreover, the larger the pores are, the greater the amount of mercury trapped inside them (Wang et al., 2022). Mercury compression testing involves expensive instrumentation and is labor-intensive and time-consuming. Furthermore, mercury is highly volatile and toxic, posing a potential risk in contaminated laboratories. After mercury pressure testing, mercury-contaminated samples require centralized disposal.

Similar to MIPs, GA operations rely on the principle that physical adsorption occurs at low temperatures on the surface of a sample placed in a gas system. Thus, the adsorption and desorption isotherms of the gases can be obtained. Through subsequent analysis, parameters such as pore morphology, pore-specific surface area, pore size distribution, and pore volume can be determined (Zhang & Kong, 2013). Nitrogen is commonly used as the adsorbate. However, owing to limitations in the sample size and manometer accuracy, nitrogen adsorption (NA) may overlook the presence of macropores. Unlike MIP, which often targets connected macropores, NA primarily characterizes micropores and mesopores. Consequently, it is often used alongside MIP to provide a more comprehensive qualitative and quantitative characterization of soil pore space (Zhu et al., 2019). MIP and GA exhibit similar limitations. Their pore analyses are based on simplified assumptions: pores are treated as cylindrical, and the measured pore radius represents the equivalent radius. This does not accurately or intuitively reflect the spatial structure of soil pores (Zhou, Lv & Li, 2009; Yang et al., 2013a; Yang et al., 2013b).

Nuclear magnetic resonance

Nuclear magnetic resonance (NMR) involves the resonance jump of atomic nuclei between energy levels under the influence of an external magnetic field. The distribution of soil pore throats is quantitatively characterized by acquiring the NMR relaxation signals of fluid hydrogen in the pore spaces of the samples. Subsequently, the parameters related to the pore structure and pore size are calculated and determined. One key constant in NMR analysis is transverse relaxation time T2. This time constant describes the recovery process of the transverse component of the nuclear magnetization intensity. The T2 distribution curve is particularly important for analyzing the macropore structure of the soil. By inverting the T2 distribution curve, the size and distribution of macropores within the soil can be determined (Gao et al., 2022). Kong et al. (2020) utilized NMR to obtain pore-size data from soil samples, as displayed in Fig. 2B. Pores with equivalent diameters ranging from 0.01–1 µm are primarily responsible for the porosity of the sample. Furthermore, Chu et al. (2022) applied NMR to investigate the pore damage law and generated T2 distribution curves and NMR pseudo-color images of the samples (Fig. 2C). NMR has a broad testing range and rapid speed, making it a common tool for studying the pore size distribution of soil under freeze-thaw conditions (Li et al., 2016). When compared over other techniques, such as SO, MIP, and GA, NMR offers several advantages. Not only can it determine the pore size and pore size distribution through the T2 distribution curve but it can also visualize the soil pore space in three dimensions using pseudo-color images. Furthermore, NMR allows for the quantification and estimation of water content in the soil as well as its spatial variability (Tian et al., 2014; Ma et al., 2020). This technique also permits observation of the infiltration and distribution of different solutes within the soil (Mantle, Sederman & Gladden, 2001).

Figure 2 Nuclear magnetic resonance (NMR) instrument and related images.

(A) NMR instrument (Kong et al., 2020); (B) NMR pore size distribution (Kong et al., 2020); (C) NMR pseudocolor images (Chu et al., 2022). The color bars in (C) indicate the relative intensity range of the fluid containing 1H within the pores, with black bars indicating the absence of an imaging signal. In contrast, red indicates higher fluid content within the pores and a more intense NMR imaging signal. Source credit (C) Yapei Chu.

In the NMR studies of soil samples, the transverse surface relaxation rate and pore shape factor are essential parameters for obtaining pore diameter variations from the NMR T2 spectrum. However, these parameters cannot be directly measured in practical applications, which is a significant limitation of NMR. To address this issue, Yang, Li & Nian (2023) developed a method that relied on scaled NMR T2 spectral curves obtained from MIP data. This approach enables the accurate determination of the NMR T2 spectra of soil samples, along with the pore radius conversion coefficient. Novotny et al. (2023) introduced diffusion in the internal field method, which eliminates the need to know the transverse surface relaxation rate. This method can successfully determine the pore size distribution of undisturbed soil samples and allow the estimation of water retention curves based on the pore size distribution. In addition, NMR spectroscopy is expensive. Owing to its fundamental principle, NMR is restricted to detecting the pore space occupied by a saturated medium. Prior to testing, samples require extensive pretreatment, including size adjustment, drying, vacuuming, and water saturation. This arduous process can potentially cause damage and alter the soil pore structure, preventing accurate characterization of the in situ pore structure. Consequently, NMR is not widely used for routine soil pore characterization.

Non-intrusive techniques

Compared to invasive techniques, noninvasive techniques are advantageous because they are nondestructive and provide realistic images that accurately represent the geometry and topology of the medium while preserving the original soil pore structure (Luo & Lin, 2009). Commonly utilized non-invasive techniques include SEM and X-ray CT.

Scanning electron microscopy

Scanning electron microscopy (SEM), a form of observation between transmission electron and optical microscopy, utilizes a focused high-energy electron beam to scan and image the surface of a specimen point-by-point. The interaction between the beam and material stimulates various physical information that is then collected, amplified, and re-imaged to characterize the microscopic morphology of the material. The primary imaging signals of SEM are secondary electrons. SEM offers several advantages, including a high resolution of 3–6 nm, a large depth of field, a strong sense of image stereoscopy, and a large adjustable range of magnification (100–200,000). It can also be used for various other analytical functions. In practical applications, it is often used to observe the microscopic morphologies of materials. Using SEM, researchers can observe the microparticle morphology and pore structure of soil samples on the micron or even submicron scale (Al-Mukhtar, Khattab & Alcover, 2012). SEM images are analyzed using image-processing software such as Image-Pro Plus (IPP). The software facilitates the easy acquisition of several two-dimensional pore parameters, including porosity, number, area, perimeter, equivalent diameter, shape coefficient, and anisotropy (Tang et al., 2012; Li & Li, 2017; Li et al., 2019a; Li et al., 2019b). Furthermore, the fractal dimensions could be derived using additional calculations (Liu et al., 2023). To investigate the impact of depth on the microstructure of loess, Li & Li (2017) utilized IPP to postprocess SEM images and segmented the pores based on their diameter. The results indicated that as the soil depth increased, pore connectivity became weaker, the areas of macropores and mesopores decreased, and the areas of small pores and micropores increased.

SEM is in exceptionally high demand for such samples. Before testing, the samples must undergo a series of preprocessing steps, including drying and gold spraying. During the dewatering and drying of the soil samples, there is a risk of shrinkage and deformation, making it challenging to maintain the original structural form. Additionally, owing to the random nature of SEM imaging, the sample must be scanned from left to right and from top to bottom, requiring meticulous operational skills. Furthermore, the sample size for SEM is small, typically less than 10 mm ×10 mm ×10 mm, necessitating the selection of appropriately magnified images for qualitative or quantitative analyses. If the magnification is too high, the number of observable pores diminishes, reducing the representative value and significance of the microscale guidance (Zhang & Kong, 2013). Post-processing using image processing software, such as IPP, cannot completely eliminate human error and image noise, which can also impact pore identification. The key limitation of SEM is its inability to probe and visualize the characteristics beneath the surface or within a three-dimensional body, as it only allows the observation of the surface morphology of the sample.

Computed tomography

CT, which is a widely used diagnostic tool in the medical field, has recently been adopted in soil science applications (Taina, Heck & Elliot, 2008). This advancement allows for the nondestructive, quantitative characterization of the pore structure of in situ soils. The underlying principle involves placing soil samples in a scanning chamber and utilizing X-rays to scan each layer to obtain cross-sectional samples. Macropore extraction relies on the distinct X-ray absorption capacity of media of various densities within the soil (Figs. 3A, 3B). Compared with other techniques, the primary advantage of CT is its ability to generate continuous images reflecting the pore structure of the soil without any preprocessing. This capability offers a true representation of the three-dimensional structure of soil pores, ensuring nondestructive and accurate characterization (Cheng, Liu & Zhang, 2012; Wang, Qian & Gao, 2021). By combining multiple two-dimensional slices obtained from scanning with image processing software such as Avizo, VG Studio MAX, MATLAB, ImageJ, MAVI, Blob3D, and Fiji, the three-dimensional spatial structure of a sample can be reconstructed (Wen, Chen & Shao, 2022). This approach reveals a wealth of information regarding the three-dimensional characteristics of the soil structure, including the pore connectivity, sphericity, and fractal dimensions (Figs. 3C, 3D, 3E). Currently, CT is utilized to investigate how various land-use types (Wang, Qin & Bai, 2018), vegetation types (Kan et al., 2023), tillage practices (Wang, Qian & Gao, 2021), soil improvement measures (Yang et al., 2013a; Yang et al., 2013b), and reclamation age (Cai et al., 2018) impact soil pore characteristics.

Figure 3 Computed tomography (CT) and three-dimensional structural characterization of soil pores.

(A) Industrial CT scanning equipment; (B) CT scanning slices of soil samples; (C) three-dimensional reconstruction of soil samples; (D) threshold segmentation used to separate pores; (E) extraction of pore skeletons.

Currently, CT is only suitable for high-clarity testing in a controlled laboratory environment and cannot be used to conduct nondestructive in situ scanning of soil directly in the field. When performing CT, it is necessary to select an appropriate sample size and resolution for pore identification. The scanning resolution of CT is inversely proportional to the maximum size of the sample, and image resolution plays a decisive role in pore quantification (Li et al., 2019a; Li et al., 2019b). Therefore, the selection of the sample size is crucial. A low resolution may result in small pores and inactive voids being overlooked, leading to inaccurate calculations of parameters such as total porosity and connectivity. Although reducing the sample size can improve the scanning resolution, the results obtained may be less representative because of the strong non-homogeneity of the soil space and the inability of small samples to contain large pores (Cheng, Liu & Zhang, 2012). In the image-processing stage, CT has the clear disadvantage of involving a large number of cumbersome and time-consuming image analyses. Improving the scanning accuracy blindly can increase the amount of data, leading to greater difficulties in postprocessing. The most significant challenge in analyzing CT images is the effective segmentation and extraction of pores. A low scanning resolution can result in poor contrast at the solid/porous interface of CT images, potentially leading to errors in contact position judgments. Although medical and industrial CT share the same underlying physical principles, their distinct applications require distinct system layouts and designs, resulting in different resolutions and sample size requirements. Compared to industrial CT, medical CT is less expensive and allows larger sample sizes to be analyzed; however, it exhibits relatively low resolution. Currently, medical CT has a resolution of approximately 600 µm, whereas industrial CT ranges from 0.5–100 µm (Liu et al., 2016; Yi et al., 2020). Researchers can select the appropriate CT based on the sample size, pore characterization requirements, and financial budgets.

Summary of techniques

Based on the above discussion, we summarize the advantages and disadvantages of the various techniques outlined in Table 4. To be considered ideal, a soil macropore characterization technique should possess the following attributes: overallness, non-destructiveness, realism, visualizability, diversity of indicators, quantitativeness, broad measurable pore-size ranges, and low sample requirements. SO is straightforward and intuitive; however, it fails to provide a three-dimensional, nondestructive, and visual characterization of soil macropores. MIP can characterize a broad range of pore sizes, but SWRC, MIP and NA oversimplify the pore shape and lack realism in their characterization (Zhang & Kong, 2013). NMR enables three-dimensional visualization; however, it can only detect pore spaces occupied by saturated media, necessitating intricate preprocessing. SEM can only capture two-dimensional pore characteristics on the surface of soil samples, thus failing to provide three-dimensional pore characteristics within the soil body. In contrast, CT has significant advantages (Cai et al., 2023). It enables the characterization of the soil macropore structure from macroscale to microscale, from whole to local, from indirect to direct, from two-dimensional observation to three-dimensional visualization, from qualitative to quantitative, and from destructive to nondestructive. Owing to its nondestructive nature, high precision, and accessible information, CT has broad applications and prospects for soil pore characterization.

Table 4 Pros and cons of various techniques.

	Staining observation	Soil water retention curve	Mercury intrusion porosimetry	Gas adsorption	Nuclear magnetic resonance	Scanning electron microscopy	Computed tomography	
Overallness		✓	✓	✓	✓		✓	
Non-destructiveness					✓	✓	✓	
Realism	✓				✓	✓	✓	
Visualizability					✓		✓	
Diversity of indicators	✓				✓	✓	✓	
Quantitativeness	✓	✓	✓	✓	✓	✓	✓	
Broad measurable pore size ranges		✓	✓		✓	✓	✓	
Low sample requirements	✓	✓					✓	
Notes.

Overallness whether the sample is treated as a three-dimensional whole

Non-destructiveness whether the process is destructive to the sample

Realism whether the characterization is based on reality (whether pore shapes are simplified)

Visualizability whether the internal pore structure of the sample can be visualized

Diversity of indicators whether the obtained pore index is sufficiently enough

Quantitativeness whether the characterization lead to quantitative results

Broad measurable pore size ranges whether the pores in a wide range of pore sizes can be identified

Low sample requirements whether complex pre-processing is required

Furthermore, numerous researchers have discovered that integrating other techniques with CT can lead to a more intuitive, precise, and comprehensive characterization of soil macropores. For example, by combining CT with SO, it is possible to visualize the three-dimensional structure of the soil macropore space and the morphology of macropore flow simultaneously (Perret et al., 2000; Yi et al., 2020; Jia et al., 2022; Zhang et al., 2022a). The integration of CT with MIP and GA allows for the comprehensive and accurate quantification of soil porosity and pore size distribution across a wide range (Zhang & Kong, 2013; Hashemi et al., 2015; Zong et al., 2015; Zhang et al., 2020). Therefore, in the future, we can draw inspiration from this research approach and combine CT with other techniques to explore and identify pore-structure features with increased precision.

Bibliometric Analysis of Research on CT Characterization of Soil Macropores

VOSviewer (The Centre for Science and Technology Studies, Leiden, Netherlands) and CiteSpace (Drexel University, Philadelphia, PA, USA) are Java-based bibliometric analysis software tools used to construct and visualize networks of scientific measurements (Chen, 2004; Van Eck & Waltman, 2010). These tools enable users to gain a quick understanding of the knowledge and research in a specific field. In this study, we selected literature available in the WOS core collection as our research object. On July 5, 2023, we conducted a comprehensive search of literature from all potential publication years using the search formula “(((TS =(Soil-Pore)) OR (TS =(Soil-Macropore))) AND ((TS =(CT)) OR (TS =(Computed- Tomography)))) NOT (SILOID = =(“PPRN”)).” After excluding six review articles, we obtained a total of 312 relevant studies. Subsequently, we manually filtered 33 articles that were not pertinent to our study or were poorly defined through individual readings of their abstracts and full texts. This exercise resulted in a final set of 279 articles published between 1998 and 2023. To analyze the annual number distribution and keyword clustering of these research articles, we utilized the bibliometric analysis software VOSviewer and CiteSpace.

As the studies were identified using the aforementioned search formula, the search terms employed in the formula constituted the largest nodes and the most prominent emergent terms in the maps. To provide a more scientific and intuitive representation of the current status and hotspots of soil macroporosity research utilizing CT from 1998 to 2023, all maps were drawn after excluding relevant search terms such as “CT”, “Soil-Pore”, and “Macropore.”

Annual number distribution of research articles

The number of research articles on CT characterization of soil macroporosity from 1998 to 2023 was counted based on the articles retrieved in WOS. The line graph of the annual number of articles was plotted and is presented in Fig. 4. Overall, the number of research articles published each year has been increasing, and this growth can be divided into three distinct stages based on the year of publication:

Figure 4 Annual number distribution of research articles, 1998–2023.

1. 1998–2007: During this period, the number of published articles was relatively low, indicating that research in this area was still in its early stages of development.

2. 2008–2017: The number of published articles showed a steady upward trend during this period, indicating that research in this field had begun to gain momentum and entered a preheating stage.

3. 2018–2023: Research in this field began to accelerate rapidly after 2018. The average number of research articles published each year was 29.17, accounting for 62.72% of the total number of articles issued during the entire research period. This suggests that research interest and activity in this field are rapidly increasing and that they have entered a heating stage.

In conclusion, the number of research articles on the CT characterization of soil macroporosity has been increasing over time, with three distinct stages of growth based on the year of publication. The most recent stage has seen a significant acceleration in research output, indicating rapidly growing interest and activity in this field.

Analysis of research keywords

Research hotspots

In Fig. 5, VOSviewer was used to visualize the co-occurrence density of keywords at this stage. Table 5 lists the top 30 high-frequency keywords from the articles at this stage. Combining Fig. 5 and Table 5, the main research hotspots in this field are centered around “quantification”, “organic-matter”, “soil structure”, “physical-properties”, “tillage”, “image analysis”, “preferential flow”, “transport”, “solute transport”, “management”, “pore structure”, and other related topics. Among the 279 articles, “quantification” appears most frequently, with a total of 64 occurrences, followed by “organic-matter” with 49 occurrences.

Figure 5 Visualization map of keyword co-occurrence density.

Font size and density (background color) of keywords are used to represent the frequency. Greater font size indicates greater frequency.

Table 5 High-frequency keywords and frequencies for computed tomography characterization of soil macropore studies.

Keywords	Frequencies	Keywords	Frequencies	
Quantification	64	Water	23	
Organic-matter	49	Flow	22	
Soil structure	43	Macroporosity	21	
Physical-properties	38	Water-flow	21	
Tillage	38	Compaction	20	
Image analysis	37	Fractal dimension	19	
Preferential flow	35	Porous-media	19	
Transport	32	Impact	18	
Solute transport	30	Parameters	18	
Management	28	Roots	18	
Pore structure	28	System	18	
Hydraulic conductivity	27	Pore size distribution	17	
Porosity	26	Hydraulic-properties	16	
Carbon	23	Infiltration	16	
Dynamics	23	Model	16	

Figure 6 shows the clustering density of keywords in the research articles on the CT characterization of soil macropores using VOSviewer. The tool clusters related topics within the research domain and indicates their relevance among these topics. In the mapping, distinct colors represent the categories assigned to each topic, whereas the size of the topic labels reflects their frequency and occurrence. Larger labels indicate higher frequencies. The distance between topics signifies their similarity; topics that are closer share more similarities. Notably, 92 keywords appeared more than six times. VOSviewer categorizes these keywords into four clusters, purple, blue, green, and yellow, revealing four distinct research fields based on their relevance. For instance, the purple cluster (Group 1) focuses on analyzing the impact of soil pore space on “water” through “image analysis.” This cluster highlights processes such as water flow and pollutant transport, which can be influenced by various macropore properties. The keywords in this group include “preferential flow”, “solute transport”, “hydraulic conductivity”, “water-flow”, and “hydraulic conductivity”. The blue cluster (Group 2) emphasizes CT-characterizable pore properties such as “physical-properties”, “macroporosity”, “fractal dimension”, “pore size contribution”, “bulk density”, and “macropore networks”. Groups 3 and 4—the green and yellow clusters—primarily explore factors that impact soil pore structure, including “soil structure”, “tillage” practices, “vegetation” and “root” systems, land use, and “management” practices.

Figure 6 Co-occurrence network of high-frequency keywords for the characterization of soil macropores using computed tomography, 1998–2020.

Distinct colors represent the categories assigned to each topic, whereas the size of the topic labels reflects their frequency and occurrence. Larger labels indicate higher frequencies. The distance between topics signifies their similarity; topics that are closer share more similarities.

Research frontiers and trends

The process of quantifying macroporosity using CT is gradual and meticulous. To gain a comprehensive understanding of this area, we employed mutation detection in CiteSpace literature visualization software to generate the most impactful citation burst map. Figure 7 presents a detailed look at the top 30 keywords that emerged from this analysis, highlighting the significant shifts and evolutions in research trends related to the characterization of soil macroporosity using CT between 1998 and 2023. Blue indicates the emergence of keywords over time, whereas red indicates the period when these keywords experienced a surge in popularity and influence. The term “fractal dimension” is a measure that assesses the self-similarity and quantitative properties of the objects being studied. It serves as an effective metric for gauging and comparing the surface roughness. Fractal dimension is a keyword that experienced the earliest and longest bursts in popularity, starting in 2003 and ending in 2014. This suggests that fractal dimension is a crucial metric for quantifying soil macropore structure. The terms “macroporosity” and “computed tomography” have long been the focus of attention, as indicated by their extended red lines. Initially, academic research on soil macropores primarily relied on traditional methods and indicators such as “model”, “bulk density”, and “soil structure.” However, starting in the 2010s, advancements in technology such as “micro-computed tomography”, “visualization”, and “3-dimensional quantification” have emerged and gained popularity. This indicates that CT scanning combined with three-dimensional visualization has become a widely used method for the comprehensive and accurate quantification of soil pores. In recent years, advancements in CT have shifted the research paradigm beyond simple model construction. Thanks to image analysis software like Avizo, it is now possible to reconstruct the pore space in three dimensions, thereby accessing a wealth of pore “parameters” within the pore network. This expansion in research has led to a deeper understanding of soil macroporosity, beyond its influence on “solute transport”, “air permeability,” and “hydraulic property”. Researchers have also explored the impacts of different soil textures, vegetation, tillage methods, and soil compaction on soil macropores. This is evident from the keywords “clay soil”, “tillage”, “root”, “soil compaction”, “impact”, and “vegetations”.

Figure 7 Strongest citation bursts map visualizing the top 30 keywords that have characterized soil macropores using computed tomography from 1998 to 2023, as identified by CiteSpace.

The color blue represents the time at which keywords first emerged, while the color red signifies the time at which they experienced a burst in citations.

The bibliometric analysis revealed another intriguing finding: the keyword coverage visualization map (Fig. 8). This map offers a glimpse into the promising areas and advancements in academia over time. The map colors keywords based on their average year of publication, with purple indicating earlier years and yellow indicating more recent years. By analyzing this map, one can gain insights into the primary trends in the characterization of soil macropores using CT. The results indicate that keywords such as “CT images”, “soil organic carbon”, “impact”, “macropore characteristics”, “macropore networks”, “pore size distribution”, and “pore connectivity” are relatively new. Furthermore, these conclusions align with those derived from the strongest citation burst map (Fig. 7).

Figure 8 Visualization map of keyword coverage.

The map colors keywords based on their average year of publication, with purple indicating earlier years and yellow indicating more recent years.

In summary, CT has vast potential for studying soil macroporosity. Currently, researchers utilize CT scanning and three-dimensional reconstruction software to visualize and analyze soil macropores in three dimensions. Sophisticated image processing techniques permit us to obtain a comprehensive understanding of the three-dimensional network structure of macropores and accurately measure crucial parameters such as porosity, pore size distribution, connectivity, and fractal dimension. Furthermore, this line of research has delved into the correlation between macroporosity characteristics and hydraulic properties as well as soil ecological functions. We also examined how soil texture, vegetation, and agricultural practices influenced the macropore structure. With the advent of higher-resolution CT equipment and the confluence of other advanced technologies, more precise detection and comprehensive in-depth characterization of soil macroporosity are expected in the future.

Conclusion and Outlook

Current techniques for analyzing soil macroporosity include SO, SWRC, MIP, GA, NMR, SEM, and CT. Each technique offers a unique perspective, operating at different scales and dimensions. However, their advantages and disadvantages are dictated by their underlying principles, leading to specific applications and research requirements. SO is a common technique used to visualize the distribution of soil macropores. This method enables the observation of macropore patterns and characteristics in the soil profile. SWRC, MIP, GA, and NMR are other techniques that can be used to analyze the size distribution and connectivity of macropores. SEM is useful for observing the shape and surface morphology of macropores. CT can be used to quantitatively analyze the soil pore structure on both two- and three-dimensional scales, allowing for a more precise understanding of macropore features by capturing the three-dimensional structure of soil macropores. However, each of these techniques has limitations in terms of accuracy and completeness. Therefore, combining multiple techniques can compensate for these limitations and provide a more comprehensive characterization of soil macropores. In practice, researchers must choose appropriate techniques based on their research objectives and experimental conditions. This article is intended to introduce and compare the advantages, disadvantages, and applicability of the aforementioned soil macropore research techniques, and as a guide for scholars who are studying soil macropores to select appropriate characterization techniques.

CT is a promising technique for the three-dimensional characterization of soil pore structure, and the combination of CT with other techniques has attracted significant attention from scholars. Bibliometric analyses indicate that the research hotspots in the field of soil macroporosity characterization by CT are primarily centered on “quantification”, “organic matter”, “soil structure”, “physical properties”, “tillage”, “image analysis”, “preferential flow”, “transport”, “solute transport”, “management”, and “pore structure”. Currently, a popular research trend involves the use of CT in conjunction with three-dimensional reconstruction software for visualization and analysis of the soil pore space. This approach enabled the extraction of macropore parameters and a comprehensive examination of the three-dimensional network structure of the pore space.

In the field of soil macropore characterization, the combination of multiple techniques has become a crucial avenue for advancing research depth and innovation. CT plays a pivotal role in the quantitative characterization of soil pore structure due to its non-destructive and three-dimensional visualization capabilities. Therefore, the organic integration of CT with other characterization techniques is of greater importance for current and future research. It is regrettable that this review has outlined the advantages and disadvantages of various independent characterization techniques, but has not fully explored the potential and advantages of the combined application of these techniques or how they work together to solve the complex problems in soil pore characterization. This is undoubtedly a significant gap in the current research, which needs to be explored and summarized by subsequent researchers.

Supplemental Information

Supplemental Information 1 High-frequency countries and frequencies for CT characterization of soil macropore studies

Supplemental Information 2 High-frequency countries for CT characterization of soil macropore studies

The authors are very thankful to everyone who participated in this study.

Additional Information and Declarations

Competing Interests

Author Contributions

Data Availability

The authors declare there are no competing interests.

Yanwen Cao conceived and designed the experiments, performed the experiments, analyzed the data, prepared figures and/or tables, and approved the final draft.

Kai Zhang conceived and designed the experiments, authored or reviewed drafts of the article, and approved the final draft.

Shuyu Liu performed the experiments, analyzed the data, prepared figures and/or tables, authored or reviewed drafts of the article, and approved the final draft.

Yilin Wang analyzed the data, prepared figures and/or tables, and approved the final draft.

The following information was supplied regarding data availability:

This is a literature review and did not utilize raw data.

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
