# Peer review of "A review of advancements in the theory and characterization of soil macropore structure"

_PeerJ, doi:10.7717/peerj.18442_

## Round 0.1 · original submission · Major Revisions

Two subject matter specialists have critically reviewed the manuscript. I suggest that the authors revise it accordingly and resubmit. Since it is a literature review, please add more recent publications in the subject area, find the research gaps, and provide recommendations based on your review.

Reviewer 1 ·

Basic reporting

• While manuscript is a literature review article and the authors claim to have included research literature from diverse regions, the manuscript seriously lacks to include important works on soil macropores, their characterization/quantification, and their functions especially their relationships with water and solute transport published by several researchers (eg. NJ Jarvis, S. Schluter, HJ Vogel, J. Koestel, L. Pires, …). Few examples include:
o Schlüter, S., Vogel, HJ. (2022). Quantification of Soil Porous Architecture. In: Jon Mooney, S., Young, I.M., Heck, R.J., Peth, S. (eds) X-ray Imaging of the Soil Porous Architecture. Springer, Cham. https://doi.org/10.1007/978-3-031-12176-0_6
o Nimmo, J. R., Katuwal, S., & Lucas, M. (2023). Porosity and pore-size distribution. In M. J. Goss & M. Oliver (Eds.), Encyclopedia of soils in the environment (second edition) (pp. 16-24). Oxford: Academic Press.
o Jarvis NJ. A review of non-equilibrium water flow and solute transport in soil macropores: principles, controlling factors and consequences for water quality. Eur J Soil Sci. 2020; 71: 279–302. https://doi.org/10.1111/ejss.12973
It is difficult to comprehend how the submitted work provides additional relevant information to the readers than the review article by Jarvis (2020) and the book chapter by Nimmo et al. (2023) mentioned above.

• The manuscript describes different methods that have been used in characterization of soil macropores, compares and contrasts them. With all these methods described, one of the most common and well-established methods to characterize soil pores and pore-size distribution, the soil water retention curve” is not included in the manuscript while the information included in much of the definition (which in very loosely defined) and classification of macropores rely on the studies based on X-ray CT scanning where the size of pores that can be visualized and quantified using X-ray CT is dependent on the image resolution which in turn depends on the machine specifications, size of soil sample, and the configuration of the sample within the equipment.

Experimental design

• As stated in "Basic reporting" the manuscript relies heavily on X-ray CT works and does not adequately represent/report some of the important works related soil macropores, their characterization/quantification, and their functions especially their relationships with water and solute transport published by several researchers.

• A substantial part (almost half) of the manuscript describes how the review was conducted, which could have been easily described in 2 to 3 paragraphs and does not add much value to the manuscript. Table 4 and Figures 5-10 do not provide any additional value to the potential readers.

• While the manuscript is supposed to provide information on the characterization of macropores, the manuscript merely lists a number of indicators describing macropores without describing what the indicators mean, how they are calculated, or how they related to the functional behavior of the macropores.

Validity of the findings

Please see the comments for "Basic reporting" and "Study design".

Reviewer 2 ·

Basic reporting

All my comments are contained in the pdf document attached below.

Experimental design

All my comments are contained in the pdf document attached below.

Validity of the findings

All my comments are contained in the pdf document attached below.

Annotated reviews are not available for download in order to protect the identity of reviewers who chose to remain anonymous.

---

## Round 0.2 · Minor Revisions

Please revise and resubmit. Pay special attention to pore size classification, and relevant literature. Also, try to restructure the manuscript itself for good. As the reviewer suggested, a well-structured review article is always effective.

Reviewer 3 ·

Basic reporting

Line 38- Soil pore definition is confusing
Line 39- Soil pores classification into different categories but only 1 category is defined i.e., macropores. Either define other pores, micro and meso or modify the line to accommodate the importance of macropore studies in solute transport.
Line 51-52- No citation for changing macropore shape idea
Introduction uses soil pore ecological function a lot but it is not explicitly defined what do the authors mean by that
Definition and classification of macropores
Lines 116 and 117 state that the academic community has classified macropores on the basis of equivalent diameter (ED), capillary potential, and function, however the paragraphs leading to this statement classifies macropores on the basis of size, origin, morphology, etc. a consistent flow in terminologies or definitions would be helpful for the reader and not get lost.
Since this is a review paper, a detailed review of all the macropore categories, along with their citations, would be beneficial. Also, a well-structured review would be effective.
Line 161: The previous line is in present tense whereas this line is in the past “were”
Factors affecting macropore structure
Again the paragraph could be well structured. Different factors are presented randomly and could be well-defined.

The subheadings of the Definition and classification of macropores, indicators describing the nature of macropores mostly define similar things in terms of types of macropores or classification of macropores. These can either be combined into one or remove the repeated content.
The indicators defined are a part of pore morphological properties.

Characterization techniques for soil macropore structure
Line 233: Better to explicitly write what is SO
Line 237, 238 The larger the pore space is, the deeper the staining: This line does not make sense. I think it should be more connected the pore space in depth, deeper the staining
Line 261 SWRC, MIP and GA write full forms?
Figure 1: (C) could be improved in resolution

Experimental design

no comment

Validity of the findings

no comment

---

## Round 0.3 · accepted · Accept

Please change the title to "A review of the advancements in the theory and characterization of soil macropore structure".

Reviewer 3 ·

Basic reporting

The authors have improved the general outline of the article based on the suggestions.

Experimental design

The authors have made the suggested changes

Validity of the findings

This review article has summarized most of the work done in the soil pore characterization field. It will be beneficial to the scientific community involved with the contaminant transport in the soil profile due to the presence of large macropore connectivity.